# Optimising long-term athletic development: An investigation of practitioners' knowledge, adherence, practices and challenges

**Kevin Till**[1,2]*, **Rhodri S. Lloyd**[3,4,5], **Sam McCormack**[1,6], **Graham Williams**[7], **Joseph Baker**[8], **Joey C. Eisenmann**[1]

1 Carnegie School of Sport, Leeds Beckett University, Leeds, United Kingdom, 2 Leeds Rhinos Rugby League Club, Leeds, United Kingdom, 3 Youth Physical Development Centre, Cardiff School of Sport, Cardiff Metropolitan University, Cardiff, United Kingdom, 4 Sports Performance Research Institute New Zealand (SPRINZ), AUT University, Auckland, New Zealand, 5 Centre for Sport Science and Human Performance, Waikato Institute of Technology, Hamilton, New Zealand, 6 England Performance Unit, Rugby Football League, Leeds, United Kingdom, 7 Millfield School, Somerset, United Kingdom, 8 School of Kinesiology and Health Science, York University, Toronto, Ontario, Canada

* K.Till@leedsbeckett.ac.uk

**Data Availability Statement:** All survey results are available in Leeds Beckett University repository - https://eprints.leedsbeckett.ac.uk/id/eprint/7628/.

## Abstract

Long-term athletic development practices have been recommended for the past two decades. However, limited research exists exploring the knowledge and skills required by practitioners to optimise long-term athletic development. Therefore, this study aimed to evaluate the knowledge, adherence, practices, and challenges of practitioners responsible for delivering long-term athletic development. A mixed methods survey was completed by 236 practitioners (e.g., sport coaches, physical education teachers) consisting of four parts; 1) demographics, 2) knowledge, 3) adherence, and 4) practices and challenges. Quantitative and qualitative data were analysed by Friedman's analysis of variance and thematic analyses, respectively. Quantitative findings showed practitioners 1) recognised their responsibility for delivering long-term athletic development, 2) have a familiarity with existing developmental models, and 3) had high adherence, focused upon health and wellbeing, to delivering long-term athletic development. However, practices associated with growth and maturity, monitoring and assessment, and the systematic progression and individualisation of training had lower adherence. Qualitative analysis indicated that practitioner's perceived definitions of athleticism and long-term athletic development were inconsistent, especially according to the psychological components (i.e., confidence). Practitioners' descriptions of their long-term athletic development practices identified two higher order themes; 1) goals, in which long-term athletic development "is for life" and the importance of "an individual centered journey" highlighted as sub-themes; and 2) realities of delivering long-term athletic development, whereby variety in programme delivery, monitoring development and practical challenges were noted as key priorities. Eight practical challenges were identified including governance and priorities, resources, education, early specialization, high training volumes, staff communication, parents and youth motivation. This mixed method survey highlighted a multitude of knowledge, adherence, practices and challenges towards long-term athletic development. These novel findings can help inform policy to optimise long-term athletic

**Funding:** The authors received no specific funding for this work.

**Competing interests:** The authors have declared that no competing interests exist.

development and to support the complex problem of developing a healthier, fitter and more physically active youth population.

## Introduction

Youth sport pathways often have multiple goals, ranging from maximising health, fitness and physical activity [1] to creating developmental opportunities for the potential sporting superstars of tomorrow [2]. Over the last two decades, several development models have been formalised, established and implemented, predominantly focusing upon the progression of sporting talent (e.g., Long-Term Athlete Development [LTAD] model, [3]; Developmental Model of Sports Participation [DMSP], [4]; Foundations, Talent, Elite and Mastery [FTEM] framework, [5]). However, recent consensus [6] and position [7] statements have questioned the 'talent only' strategy, especially considering the large number of youth who experience recreational and competitive sport compared to elite sport and the requirement to maximise health, fitness and physical activity for all youth. At the same time, a series of literature reviews on the concept of long-term athletic development [1,8] culminated in the publication of the National Strength and Conditioning Association (NSCA) position statement [7]. The concept of long-term athletic development was defined as the "*habitual development of athleticism over time to improve health and fitness, enhance physical performance, reduce the relative risk of injury, and develop the confidence and competence of all youth*" ([7] p. 1492). The NSCA position statement aimed to (a) help foster a more unified and holistic approach to long-term athletic development, (b) promote the benefits of a lifetime of healthy physical activity, and (c) prevent and/or minimize sport and physical activity-related injuries for all boys and girls.

A driving factor of long-term athletic development is to enhance athleticism, which is defined as the "*ability to repeatedly perform a range of movements with precision and confidence in a variety of environments, which require competent levels of motor skills, strength, power, speed, agility, balance, coordination, and endurance*" ([7], p. 1491). Both long-term athletic development and athleticism reflect interdisciplinary concepts important for health, physical activity, and sports performance, which consider the physical, motor skill and psychosocial factors (e.g., confidence) associated with youth development. To support practitioners, the NSCA position statement proposed ten pillars for successful long-term athletic development (see Table 1; [7]) based upon previous models [4] and up to date research evidence (e.g., [6,9–12]). Whilst the NSCA position statement and recent research related to long-term athletic development has emerged within the fields of sport science, physical education, coaching, and strength and conditioning (e.g., [6,10,13,14]), it remains to be determined whether these strategies are consistently implemented in practice. For example, whilst research has evaluated the implementation of the LTAD model in Canada (e.g., [15–17]), no research to our knowledge has explored the concept of long-term athletic development. Furthermore, a paucity of practical guidance and evaluation of the knowledge and skills of practitioners to successfully implement the ten pillars of long-term athletic development currently does not exist. This message could be further exemplified when considering the declining trends in physical fitness (e.g., [18]), physical activity (e.g., [19]) and motor skill development (e.g., [20]), alongside increased overweight and obesity prevalence (e.g., [21]) within youth populations.

Therefore, while long-term athletic development is recognised as a guiding principle for youth development, to date no research to our knowledge has explored the knowledge, adherence, practices, and challenges of practitioners responsible for implementing and delivering long-term athletic development programmes to youth. In this context, 'practitioners' are recognised as individuals responsible for the athletic development of youth including sport

**Table 1. Ten pillars for successful long-term athletic development [7].**

| Pillar | Description |
|---|---|
| 1 | Long-term athletic development pathways should accommodate for the highly individualized and non-linear nature of the growth and development of youth. |
| 2 | Youth of all ages, abilities and aspirations should engage in long-term athletic development programs that promote both physical fitness and psychosocial wellbeing. |
| 3 | All youth should be encouraged to enhance physical fitness from early childhood, with a primary focus on motor skill and muscular strength development. |
| 4 | Long-term athletic development pathways should encourage an early sampling approach for youth that promotes and enhances a broad range of motor skills. |
| 5 | Health and wellbeing of the child should always be the central tenet of long-term athletic development programs. |
| 6 | Youth should participate in physical conditioning that helps reduce the risk of injury to ensure their on-going participation in long-term athletic development programs. |
| 7 | Long-term athletic development programs should provide all youth with a range of training modes to enhance both health- and skill-related components of fitness. |
| 8 | Practitioners should use relevant monitoring and assessment tools as part of a long-term athletic development strategy. |
| 9 | Practitioners working with youth should systematically progress and individualize training programs for successful long-term athletic development. |
| 10 | Qualified professionals and sound pedagogical approaches are fundamental to the success of long-term athletic development programs. |

coaches, sports administrators, strength and conditioning coaches, physical education (PE) teachers, physiotherapists, and other health care providers. Whilst research exploring long-term athletic development is limited, recent survey research has examined practitioners' practices within specific sports including rugby union [22] and cricket [23], and within wider issues related to youth soccer including injury prevention [24], talent identification [25] and growth, maturation and training load [26]. The purpose of this study was to evaluate the knowledge, adherence, practices, and the challenges of practitioners responsible for implementing long-term athletic development. Such information would be vital for optimising long-term athletic development strategies including supporting the theoretical and applied knowledge and skills of practitioners responsible for delivering and implementing long-term athletic development programmes alongside identifying common challenges that should be explored within future applied research.

## Materials and methods

### Study design

This exploratory study utilised an online, self-administered survey, which was adapted from previous surveys in other domains (e.g., [22,25,27]) in conjunction with the NSCA ten pillars of successful long-term athletic development [7]. The survey used a combination of closed and open-ended questions, both quantitative and qualitative in nature, to assess practitioners' knowledge, adherence, practices and challenges associated with long-term athletic development within their context. The survey was circulated internationally on a criterion-based sampling method of practitioners responsible for the long-term athletic development of youth aged 5–16 years in a role responsible for physical activity, physical education or sports performance and was available to complete for ten weeks.

## Participants

Two hundred and thirty-six practitioners (n = 204 male [86%], n = 29 female [12%]; 3 undisclosed [2%]) participated in the online survey between June and September 2020. The sample included 44 PE teachers, 43 sport coaches, 66 strength and conditioning coaches, 31 sport leaders and/or administrators, 24 academics, and 38 identified as "other" (including medical practitioners, performance analysis and coach education within their primary roles). One hundred and seventy-eight (75%) participants also identified having a secondary role across the classifications above. The average age of participants was 37.8 ± 9.8 years (range = 22–65 years), with approximately 13.8 ± 8.4 years' practical experience (ranging from 1–40 years). Participants' academic qualifications included secondary education (n = 5), Advanced Level qualifications (A-Levels) / Business and Technology Educational Council (BTEC) qualifications (n = 6), BSc (n = 60), postgraduate qualification (n = 35), MSc (n = 96), and PhD or Professional Doctorate (n = 27). Participants' coaching qualifications ranged from Level 1 to Level 4 in the United Kingdom Coaching Certificate framework (or equivalent) across multiple sports (including rugby, football, field hockey, ice hockey, handball, baseball, volleyball, weightlifting, netball, athletics, swimming, Gaelic games, ski and snowboard, gymnastics, cricket, judo, dance, multi-skills, dodgeball, racquetball, triathlon, trampolining, and golf). Seventy-six (32%) participants were accredited with a strength and conditioning association (i.e., NSCA, United Kingdom Strength and Conditioning Association, or Australian Strength and Conditioning Association). Participants identified working with individuals of both sexes (n = 164), males only (n = 62) or females only (n = 9), across multiple ages (childhood, adolescence, adults), standards (e.g., school, community, talent, performance) and sports. Institutional ethical approval was obtained from the lead author's University, and all participants were informed of the risks and benefits of the study before providing electronic informed consent form.

## Measures

Online software (Qualtrics, Provo, USA) was used to create a survey about practitioner's knowledge, adherence, practices and challenges for optimising long-term athletic development. The survey (in English only) was created by a panel of four experts (KT, JE, RSL, GW) with a history of publications in the area of long-term athletic development and extensive experience working within youth settings and with youth practitioners. The survey consisted of four sections: 1) Demographics, 2) Knowledge, 3) Adherence, and 4) Practices and Challenges. Demographics included closed and open questions on their job role, years of coaching experience, qualifications, and their participants. The knowledge section asked practitioners to identify their responsibility for long-term athletic development using multiple choice answers (Yes, No, Maybe), define athleticism and long-term athletic development with open-ended answers and identify their familiarity with four key developmental models (i.e., LTAD model, [3]; DMSP, [4]; Youth Physical Development [YPD] model, [28]; and the NSCA ten pillars of successful long-term athletic development; [7]) using a 5-point Likert scale (1 = unfamiliar, 5 = very familiar). The adherence section asked practitioners to rate how well their current coaching or programme delivered against the NSCA ten pillars. Participants rated each of the ten pillars on a 5-point Likert scale (1 = definitely not, 2 = no, 3 = not sure, 4 = yes, 5 = definitely yes). Lastly, for the practices and challenges section, participants were asked to describe their current practices and the challenges towards the NSCA ten pillars of successful long-term athletic development using open ended answers with qualitative responses required. The survey was developed, reviewed, and pilot tested for content and face validity by experienced practitioners in both long-term athletic development and applied research (>5 years; [29]). Participants were recruited and provided access to the survey through professional networks via

email and contact through social media platforms. Surveys that were at least 50% completed were included in the analysis.

## Data analyses

As the survey contained multiple fixed-response and open-ended questions, multiple data analysis methods were used. The survey responses were initially collected and analysed using Microsoft Excel and SPSS 26.0 (SPSS Inc, Chicago, USA). Fixed response, quantitative data were reported as medians and interquartile ranges (IQR), deviations or frequencies and percentages of total responses where appropriate. Due to the non-parametric data, where differences were compared (i.e., between role, development models and adherence to the ten pillars) a Friedmans analysis of variance (ANOVA) was used with the Wilcoxon signed-rank test and Bonferroni post-hoc correction. For the open-ended questions for defining long-term athletic development and athleticism, participants' answers were analysed according to qualitative content analysis [30]. For current practices and challenges towards long-term athletic development, the qualitative open-ended questions were analysed using thematic analysis [31]. This method was chosen as it allowed practices and challenges of the practitioners to be understood. First, the answers to these questions were read on numerous occasions to allow the researcher to become familiar with the data followed by codes being generated inductively from the practitioners' answers to highlight and label the primary aspects of the transcripts. Next, codes were placed into themes, which identified the main concepts and were reviewed regularly to ensure all data and relevant information were collected. This resulted in the emergence of lower order themes which were categorised into sub-themes and higher order meta-themes [32]. The final two stages involved naming and defining the themes and providing descriptors of each theme. This was primarily carried out by the lead researcher who engaged with the research team in constant discussion and a critical friend (experienced in both long-term athletic development and applied research in practice).

## Results

### Knowledge: Responsibility for long-term athletic development

Overall, practitioners stated they were primarily responsible for the development of health and fitness (Yes = 96.1%, Maybe = 3.0%, No = 0.9%), physical performance (Yes = 91.4%, Maybe = 7.3%, No = 1.3%), confidence and competence in a range of movements (Yes = 97.0%, Maybe = 1.3%, No = 1.7%), and reducing the relative risk of injury (Yes = 93.5%, Maybe = 6.0%, No = 0.5%).

### Knowledge: Definition of athleticism

Practitioners' definitions of athleticism were analysed according to four key concepts; 1) movement ability, 2) physical qualities, 3) psychological components, and 4) multiple environments (according to [7]). The count and percentage of practitioners (with example definition) who defined athleticism according to these four criteria are displayed in Table 2. The percentage of practitioners who identified movement, physical qualities, multiple environments, and psychological components within their athleticism definitions was 74.0%, 54.3%, 36.5% and 2.0% respectively.

### Knowledge: Definition of long-term athletic development

Table 3 presents the content analysis of practitioners' definitions of long-term athletic development. Practitioners included the terms "development", "long-term", "participants (individual

**Table 2. Practitioners definition of athleticism according to four concepts; movement, physical quality, psychological, and multiple environments.**

| No. of concepts in definition | Count (%) of Respondents | Example Practitioner Definition |
|---|---|---|
| 0 | (2) 0.9% | *"of vital importance to excel further"* |
| 1 | (67) 30.6% | *"the ability to move through a wide range of fundamental and functional movement skills"* |
| 2 | (111) 50.7% | *"being able to manage their bodies and move efficiently in a variety of situations"* |
| 3 | (33) 15.1% | *"one's ability to perform a broad range of movements, with confidence and competence, to develop a range of physical qualities including strength, speed, power, agility and endurance"* |
| 4 | (3) 2.7% | *"the ability to participate confidently and competently in a range of physical activities and in doing so demonstrate flow, strength, speed and coordinated range of motion"* |

or athlete)", "the how", and "outcomes" within their definitions of long-term athletic development.

## Knowledge: Familiarity of developmental models

Table 4 presents the median (IQR) for practitioners' knowledge and familiarity of four key development models. Friedman's ANOVA showed overall significant differences between the models for all practitioners ($X^2(2) = 100.4$, $p<0.001$), academic ($X^2(2) = 16.9$, $p = 0.001$), strength and conditioning coach ($X^2(2) = 87.7$, $p<0.001$), sport coach ($X^2(2) = 33.5$, $p<0.001$), sport leader ($X^2(2) = 23.8$, $p<0.001$) and other ($X^2(2) = 14.3$, $p = 0.003$). There was no significant difference identified between the models for PE teachers ($X^2(2) = 5.6$, $p = 0.13$). Pairwise comparisons are shown in Table 4 but overall, the LTAD and YPD model were ranked with higher familiarity than the DMSP and NSCA ten pillars. When compared by role, significant differences ($p<0.001$) were found between groups for each model. Pairwise comparisons identified that sports leaders, academics, and strength and conditioning coaches had the greatest

**Table 3. Themes associated with practitioners definition of long-term athletic development.**

| Theme | Count (%) of Respondents | Explanation | Example Data |
|---|---|---|---|
| Development | 148 (67.2%) | Practitioners identified that long-term athletic development was a development process associated with developing athleticism (n = 86, 39.1%) or holistic (i.e., range of technical, tactical, physical and psychological) skills (n = 61, 27.7%) | *"the process of developing and nurturing athletes throughout their lifespan, with the long-term aspect meaning there is a systematic approach"* |
| Long-Term | 135 (61.4%) | Practitioners identified long-term athletic development was a long-term or lifelong process | *"Taking a long-term view of the work they're doing. No quick fixes. Big picture"* |
| Participants | 124 (56.4%) | Practitioners identified 'a participant' was who long-term athletic development was aimed at. Practitioners stated this could be an athlete(s) (n = 53, 24.1%) or individual (n = 73, 32.2%) | *"providing athletes (player and person) with the key athletic foundations required to meet and cope with the demands"* |
| The 'How' | 128 (58.2%) | Practitioners identified that there was an environmental, training and planning process towards long-term athletic development | *"When there is a clear plan in place of achievable and challenging goals or aims for an athlete to reach. This process should be under constant review from the coach and be flexible enough to change to adjust to the athlete's needs."* |
| Outcomes | 109 (49.5%) | Practitioners identified an endpoint or outcome of long-term athletic development including lifelong physical activity, health and fitness, participation within sport, maximising potential and achieving elite success | *"Developing the child's Athleticism for a lifetime of sport engagement not just youth success"* |

**Table 4. Practitioners familiarity of youth development models by primary role (median (Interquartile Range; IQR)).**

| Role | DMSP | LTAD | YPD | NSCA | Pairwise between Models |
|---|---|---|---|---|---|
| All | 3 (2) | 4 (1) | 4 (3) | 3 (2) | LTAD > YPD > DMSP, NSCA |
| Academic (A) | 4 (2) | 4 (1) | 5 (1) | 4 (2) | LTAD, YPD > DMSP, NSCA |
| Sport Coach (C) | 3 (2) | 4 (1.75) | 3 (2) | 3 (2) | LTAD > DMSP, YPD, NSCA |
| S&C Coach (SC) | 2.5 (2) | 4 (1) | 5 (1) | 4 (1) | LTAD, YPD > NSCA > DMSP |
| PE Teacher (P) | 3 (2.5) | 4 (3) | 3 (3) | 3 (3.5) | |
| Sport Leader (L) | 4 (1) | 5 (1) | 4 (2) | 3 (3) | LTAD > DMSP, YPD, NSCA; DMSP > NSCA |
| Other (O) | 3 (2.5) | 4 (2.5) | 3 (3.5) | 3 (2.5) | LTAD > DMSP, YPD, NSCA |
| **Pairwise between Role** | L > SC, P, C A > SC, O | L > P, C, O SC > P | A, L, SC > C A > O | A, SC > C, O | |

Data reported as median (IQR); Likert Scale 1–5; DMSP = Developmental Model of Sports Participation [4]; LTAD = Long-Term Athlete Development model [3]; YPD = Youth Physical Development model [28]; NSCA = National Strength & Conditioning Association 10 Pillars of successful long-term athletic development [7].

familiarity with the models, however strength and conditioning coaches had less familiarity with the DMSP.

## Adherence: Delivery against the NSCA 10 pillars of long-term athletic development

Table 5 shows the median (IQR) for practitioners' adherence of their current coaching or programme delivery against the NSCA ten pillars of successful long-term athletic development. Friedman's ANOVA showed an overall significant difference between each of the ten pillars for all practitioners ($X^2(2) = 174.4$, p<0.001), sport coaches ($X^2(2) = 34.9$, p <0.001), strength and conditioning coaches ($X^2(2) = 46.4$, p <0.001), PE teachers ($X^2(2) = 41.3$, p <0.001), sport leaders ($X^2(2) = 32.9$, p<0.001), and others ($X^2(2) = 41.2$, p<0.001). There was no overall

**Table 5. Practitioners adherence to the ten pillars of successful long-term athletic development (median (Interquartile Range; IQR)).**

| Role | Pillar 1 | Pillar 2 | Pillar 3 | Pillar 4 | Pillar 5 | Pillar 6 | Pillar 7 | Pillar 8 | Pillar 9 | Pillar 10 | Pairwise |
|---|---|---|---|---|---|---|---|---|---|---|---|
| All | 4 (0) | 4 (1) | 4 (1) | 4 (1) | 5 (1) | 4 (1) | 4 (1) | 4 (1) | 4 (1.75) | 4 (1) | P5 > All P2, P4, P6, P7, P10 > P1, P8, P9 |
| Academic | 4 (1) | 4 (0.5) | 4 (1) | 4 (1) | 5 (1) | 4 (1) | 4 (1) | 4 (1.5) | 4 (0.5) | 4 (1) | |
| Sport Coaches | 4 (1) | 4 (1.75) | 4 (1) | 4 (1.75) | 5 (1) | 4.5 (1) | 4 (1.5) | 4 (1) | 4 (1.75) | 4 (1.75) | P5 > P8, P9 |
| S&C Coaches | 4 (1) | 4 (1) | 4 (1) | 4 (1) | 5 (1) | 5 (1) | 4 (1) | 4 (0.5) | 4 (1) | 5 (1) | P5, P6 > P1, P8, P9; P10 > P8 |
| PE Teachers | 4 (1) | 4 (1) | 4 (1) | 4 (1) | 5 (0) | 4 (2) | 4 (1) | 4 (2) | 4 (3) | 4 (1) | P5 > P1, P2, P4, P8, P9 |
| Sport Leaders | 4 (1.25) | 5 (1) | 4.5 (1.25) | 4.5 (1) | 5 (0.25) | 4 (1) | 4 (1) | 4 (1) | 3 (1.25) | 5 (1) | |
| Other | 4 (1.75) | 4 (1) | 4 (2) | 4 (1.75) | 5 (1) | 4 (1) | 4 (1.5) | 4 (2) | 3.5 (2) | 4 (1.75) | P5 > P3, P8, P9 |

Data reported as median (IQR); Likert Scale 1–5; Pillar (P) P1—Long-term athletic development pathways should accommodate for the highly individualized and non-linear nature of the growth and development of youth; P2—Youth of all ages, abilities and aspirations should engage in long-term athletic development programs that promote both physical fitness and psychosocial wellbeing; P3—All youth should be encouraged to enhance physical fitness from early childhood, with a primary focus on motor skill and muscular strength development; P4—Long-term athletic development pathways should encourage an early sampling approach for youth that promotes and enhances a broad range of motor skills; P5—Health and wellbeing of the child should always be the central tenet of long-term athletic development programs; P6—Youth should participate in physical conditioning that helps reduce the risk of injury to ensure their on-going participation in long-term athletic development programs; P7—Long-term athletic development programs should provide all youth with a range of training modes to enhance both health- and skill-related components of fitness; P8—Practitioners should use relevant monitoring and assessment tools as part of a long-term athletic development strategy; P9—Practitioners working with youth should systematically progress and individualize training programs for successful long-term athletic development; P10—Qualified professionals and sound pedagogical approaches are fundamental to the success of long-term athletic development programs.

significant difference in rank by academics ($X^2(2) = 15.7$, p = 0.074). Overall, Pillar 5 (health and wellbeing) was the significantly highest scoring pillar. Conversely, Pillars 1 (highly individualized growth and development), 8 (monitoring and assessment) and 9 (systematically progress and individualize training programmes) were the significantly lowest scoring pillars. When compared by role, no significant differences were identified between practitioners.

## Practices and challenges: Practitioners perspectives

Practitioners were asked to describe their long-term athletic development practices and the challenges associated with these. Two main higher order themes were identified, which were "The goals of long-term athletic development" and "The realities of delivering long-term athletic development".

## The goals of long-term athletic development

The goals of long-term athletic development were centred around two main themes "It's for Life", and "The Individual Centred Journey".

**It's for life.**   Practitioners perceived that long-term athletic development is not just for youth but rather "*for life*" and is a lifelong process starting in childhood. Practitioners perceived long-term athletic development was "*integral*", "*essential*", "*a priority*" and "*our duty*" for all; with one PE teacher emphasising that "*If this is not our goal, we are in this profession for the wrong reasons*". Practitioners acknowledged that long-term athletic development should encourage lifelong participation within "*physical activity*", "*recreational sport*", "*competitive sport*" and "*elite sport*" summarised by these PE Teachers,

> "*Our goal should be to inspire a lifelong pursuit of physical activity whether recreationally or competitively. Long-term athletic development provides all youth those pathways if appropriately applied.*"

> "*For me it's about lifelong physical activity and engagement, not just a professional sporting career.*"

This lifelong focus on physical activity and sport was related to outcomes of increased "*health*" and "*wellbeing*". Health and wellbeing are presented within the NSCA Pillar 5, which practitioners rated the most important in relation to their current practices. This was further emphasised with PE teachers and sport coaches stating, "*it's all about the health*", and "*it's the foundation of living a healthy active life*" with "*wellbeing our highest priority with appropriate support and education embedded within our programmes*". Whilst health and wellbeing were identified as the central aspects of their practice, "*holistic development*" was another outcome that practitioners focussed upon within their long-term athletic development practices. Practitioners aimed to develop "*well rounded individuals*" and focussed upon "*better people first, then better athletes*".

**The individual centred journey.**   Practitioners identified the need for an "*individual centred*" and "*athlete centred*" approach which was summarised succinctly by a PE teacher "*The person is at the centre of everything I do. Results are secondary. Enjoyment is primary*". Multiple practitioners identified "*the child comes first*" or "*the athlete is central*" in their practices, which related to the subthemes of "health", "wellbeing" and "holistic development" identified in the "It's for Life" theme. Focussing on these outcomes means that an individual centred journey is more likely, with practitioners highlighting that they were in a "*people business*" and that they "*coach people, not sport*".

To facilitate an individual centred journey, three sub themes emerged relating to the environment needed to allow this to occur. The first was in relation to *"enjoyment"* and *"fun"* whereby *"enjoyment is always a priority over performance"*, *"athletes should enjoy the journey"* and *"there is a primary concern of fun and optimising health and wellbeing"*. Second, practices required a *"developmental"* focus whereby individuals can *"develop"*, *"improve"* and *"progress"* towards their health, fitness, sport and physical activity goals. However, some practitioners identified that sport may not always be the best means for *"development"* to occur with individuals, which aligns with Pillar 2 of the NSCA ten pillars. This offers further support that the individual is central and developing programmes aligned to the individual's interests are needed, summarised by a sports coach,

> "*In my opinion there is too much of an emphasis on developing physical competence through sport. Every child should be guided through an LTAD model regardless of interest in sport. Too many fall between the cracks and miss out on the benefits (of physical activity) during childhood and in later life*"

The third sub theme related to creating a *"safe and caring"* environment, where practitioners *"care"* about individuals and where *"social interactions"* are vital. This caring and social environment can be created by all practitioners to establish motivation, encouragement and enjoyment for sport and physical activity, summarised by a strength and conditioning coach,

> "*When players get older they won't remember what sets and reps you did with them, but they will remember how you made them feel. Were you motivating, were you encouraging? Did you care?*"

### The realities of delivering long-term athletic development

The realities of delivering long-term athletic development centred around three main themes, "The Programme—Variety is Key", "Monitoring Development", and the "Practical Challenges".

**The programme—Variety is key.**    The idea of programmes providing a variety of opportunities was identified as vital for long-term athletic development. The sub themes related to 1) multi-sport and multi-activity, 2) movement development, 3) physical development, 4) individualisation, and 5) education.

The NSCA Pillar 4 suggests an early sampling approach, which was strongly supported by most practitioners and therefore the first sub theme was "multi-sport and multi-activities". Practitioners emphasised the importance of *"variety"* and participation in a *"large range of sports and activities"*. This was identified as *"fundamental"* and a *"key"* aspect of long-term athletic development for developing *"well rounded athletes"* whilst reducing *"burnout in one sport"* as suggested by a sport leader and strength and conditioning coach,

> "*Multi sport/activity participation should be encouraged as young as possible throughout childhood. Supervised healthy risky play in several environments is appropriate such as: water, forest, land, sport, etc. in multiple planes (e.g., diving in swimming, hand stand in gymnastics).*"

> "*As children age more opportunities are usually made available to develop confidence and competence in FMS and FSS (e.g. parkour, skate boarding, karate, golf, mountain biking, wake boarding, swimming. . .)*"

To support this multi-sport and multi-activity approach, practitioners strongly *"encouraged"* youth to participate in multiple sports and activities by providing opportunities to sample *"activities within their PE curriculum"*, *"after school programmes"*, *"through play"* and *"within their sport specific sessions"*. Some practitioners suggested it was their *"responsibility"* to provide multi-sport activities where individuals specialise early, and that such programmes should be available *"until 16 years"*. However, whilst many practitioners agreed with a multi-sport approach, they acknowledged that they *"couldn't influence this"*. Practitioners suggested sampling and diversification was not *"prescribed by sports clubs"*, that *"parents, schools and clubs need to do better"*, and that some practitioners only work with children at certain ages and *"cannot be sure that this is done"*.

As suggested in the NSCA Pillar 3, the second sub theme was in relation to "movement development". Practitioners emphasised the importance of movement development and identified that *"movement"* was a *"key"* element of their programmes through the development of *"fundamental movement skills"*, *"movement competence"*, and *"movement curriculums"*. This was delivered within *"PE programmes"* and *"sport sessions"* with a focus on developing *"competent, capable and confident movers"* delivered through *"safe"*, *"progressive"*, and *"challenging"* practices that was summarised by a PE teacher and strength and conditioning coach,

> "*Our PE programme has a specific focus on movement competence and we aim to expose students to a range of different fundamental movements to challenge and progress their competence. We encourage students to be active every day and provide an extensive range of extra curricular activities.*"

> "*We have also put into place a PE program from year 3 upwards that feeds into and focuses on key movement skills, locomotion skills and stability skills.*"

Alongside movement development, the third sub theme identified was in relation to "physical development" which was associated with the NSCA Pillars 2, 3, 7 and 10. Practitioners highlighted the importance of *"physical"* and *"athletic"* development through *"physical education"*, *"sport specific"* and *"strength and conditioning"* sessions for the development of *"speed"*, *"strength"*, *"power"*, *"agility"* and *"endurance"* in conjunction with the development of movement. A popular method to implement this was through *"warm ups"* and some environments, including schools and clubs, had specialist strength and conditioning coaches to deliver these sessions demonstrated by this PE teacher,

> "*Pupils are exposed to various training modes with our school strength and conditioning and exercise programme. However, the quality of delivery in and out of the school environment may alter, therefore not all lessons within the school programme or external coaches may provide the same opportunities.*"

In reference to NSCA Pillar 9, the fourth sub theme was in relation to the "individualisation of training". Some practitioners highlighted how their practices were individualised through *"individual development plans"* and *"planning training programmes"*, whilst others suggested individualised training was a *"goal"* and an area *"to improve"*, based on an individuals' needs. These findings align with practitioners' rating of their practices whereby Pillar 9 was one of the lowest scoring pillars and has associated challenges including *"time"*, *"resources"*, *"access"* and the *"large individual differences within group delivery"*. However, strength and conditioning coaches highlighted their practices towards individualised training, especially within state schools,

"*Plans are individualised based on the technical competence and physical qualities possessed by the athlete. Physical qualities considered include their growth/maturation status, training age, strength/power/other qualities and injury risk factors.*"

"*Individualised sport specific programs that cater to the individuals needs of the athletes to help reduce injury. We create rehab programs for injuries and constantly corresponding with the physios at the school. We are about to introduce an injury tracker system at the school, so we can track the injuries across the year and then hopefully adapt future S&C programs to counteract any trends that develop.*"

The fifth sub theme was "education", *which* was identified as being delivered "*on and off the field*" around a number of areas including "*the importance of long-term athletic development*", "*physical development*", "*nutrition*", "*recovery*", and "*lifestyle*" as explained by this sports leader,

"*We offer a multi-disciplinary support programme that takes place both on and off the field. We believe that what the player does off the field is key. In turn, we attempt to educate them and their parents/guardians in a supportive/non-threatening manner. We want to help equip players with the skills they may need to be able to take responsibility for their own development now and (more so) later in life. Encouraging player ownership and education that combine staff/player driven incentives currently work well through a spiral type curriculum.*"

**Monitoring development.** In alignment with NSCA Pillar 8, the next theme was "monitoring development". Practitioners identified monitoring development as a key aspect of long-term athletic development, as "*we need to measure and manage an athlete's progression*". The common monitoring practices included "*fitness testing*", "*movement screens*", and "*growth and maturity assessments*" as suggested by this strength and conditioning coach,

"*We conducted a standardised testing battery 3 times a year for all age groups (IMTP, CMJ +DJ, 505, Sprints, 30–15 IFT). Within the testing battery, height, sitting height and body mass is collected for the calculation of maturity offset. Results from the testing battery get put onto an interactive Google document that tracks players throughout the season.*"

Furthermore, as wellbeing was identified as a key element of long-term athletic development, practitioners reported monitoring and tracking wellbeing using "*standardised questionnaires*" to "*report*" and "*present*" wellbeing data to identify and resolve any issues. To support health and wellbeing, this strength and conditioning coach said, "*we monitor and (somewhat) manage overall training loads to ensure the volume and type of training does not unnecessarily add injury risk*". However, practitioners acknowledged "*that it was difficult to get adherence to questionnaires and management of large volumes of data*". Whilst data monitoring was a common practice, other practitioners suggested that "*I don't use any monitoring or assessment tools beyond general observation*". Due to the challenges associated with monitoring including "*time*" and "*large group sizes*", practitioners emphasized the use of "*observation*" and their "*use of my eyes and ears*" and questioned the interpretation of some quantitative data as shown by this PE teacher,

"*Outside of maturation, many of these aspects can be monitored through the coach-athlete relationship (caveat being the number of athletes). Continual progress in youths makes meaningful quantitative assessment difficult to interpret. However, to create programme*

*compliance, it may be necessary to include physical tests to feedback progress from the training programme.*"

**Practical challenges.**  The third theme within the realities of delivering long-term athletic development was "practical challenges" with eight sub themes identified including "governance and priorities", "resources", "education", "early specialisation", "high training volumes", "staff communication", "parents" and "youth motivation".

One of the key practical challenges was the governance of youth sport pathways and the priorities of other practitioners' in their delivery of long-term athletic development. Some practitioners identified a *"significantly over-competitive environment"* with *"a focus of systems and programmes on particular sports"* whereby "*other sports are seen as competition"*. For example, this sports coach stated,

"*Sport in XXX sees others sports as competition. There is no collaboration for a number of reasons. The first is scheduling, games and training always clash.*"

Furthermore, responses suggested some practitioners are driven by "*personal gain"* with a focus upon "*winning"* competitions rather than individual development. Practitioners suggested some may "*disregard athletes wellbeing in order to make personal gains through achievement"* whereby individuals are not given opportunities to develop as suggested by this sport coach,

"*But many examples could be used of a lack of playing time being given to players. Deselection for games and higher-level squads is another area. Lots of coaches are more focused on their own well-being (i.e., winning games and competitions to enhance their reputation) as opposed to focusing on the wellbeing of participants.*"

A second practical challenge was the resources available to practitioners, including "*time"* and the "*number of athletes"*. Practitioners identified how implementing individualised training and monitoring is difficult within PE classes with "*classes of 30+ students, 5 times a day"* in the "*limited time frames we have with the participants"*. Furthermore, sports coaches identified how "*inconsistent attendance"* can limit the development of athleticism. However, practitioners did provide solutions to these challenges including "*providing generic progressions for groups"*, "*individual tweaks"* and using a "*being physically active at home programme"*.

The education, skills and knowledge of practitioners was a third challenge. This was evidenced by practitioners "*own acknowledgement of their lack of knowledge"*, a lack of awareness towards their responsibility for long-term athletic development (e.g., "*this responsibility rests with other departments"*; "*we have athletic development coaches to do this"*). Furthermore, others suggested that sports place "*a greater emphasis on technical/tactical development"*, which may misalign with the concept of long-term athletic development and the NSCA pillars. In some instances, physical development "*was outsourced and delivered ad hoc and was always viewed as an (annoying) extra whereas it really should have been embedded within our work"* suggesting limited alignment towards other areas of long-term athletic development.

A challenge associated with governance, priorities and education of practitioners was "*early sports specialisation within sports and schools"*. Some practitioners identified that multi-sport participation was "*not encouraged, early specialisation is"* highlighted by these sports coaches,

"*A longer term issue/challenge as some of my athletes are focused (wrongly) too much towards rugby. This is a long term culture change that will only happen if primary/secondary education focuses more on physical literacy.*"

"*School has early specialisation approach which can create problems with developing broad range of motor skills through their development.*"

Early specialisation can also occur due to youths' "*preferences*" or "*because they are compelled by coaches, parents, etc*". Whilst some sports may encourage sports sampling, there can still be a "*focus on the technical / tactical and sport specific skills*" over athleticism and psychosocial development. Such a focus can results in overuse injuries, which was identified as a concern in racket sports.

"*I have found that children who specialise in sports early in childhood, particularly unilateral sports such as racket sports, suffer from more overuse injuries than children who have not specialised from an early age. After identifying that almost 90% of children in a racket sports programme that I worked with had an injury over a 6 month period, it was a battle to convince the sport coach of the importance of more general motor skills and movements to be included in the programme even for younger age groups.*"

Whilst early sport specialisation may result in high training volumes (e.g., "*load management is a challenge in the swimming environment due to such a huge training load of up to 18–20 hrs per week*"), youth who partake in multiple sports may also be at risk of high cumulative training volumes. For example, individuals may train and compete outside of the practitioners' programme (e.g., school, social sport, club), therefore resulting in "*multiple stakeholders*" creating "*difficulty to monitor due to the multiple sources of training the athletes frequently participate in (e.g. school)*", as summarised by a PE teacher and sports coach,

"*Agree—but easier said then done. For how long? How much? What's better—athlete A who trains in 3–4 sports in every season or athlete B who focuses on 1 sport per season (i.e. fall, winter etc?). We end up contradicting ourselves by telling parents that multiple sports are better. . . all while athletes are burning out from going from sport A to sport B, all in one night.*"

"*In my current environment I feel sampling has gone too far. Participants play at least 6 different sports per year (often more). This provides a great grounding but doesn't provide time to develop high level of skill in any one sport. I would prefer a more middle of the road approach.*"

Associated with high training volumes, was the sub theme of staff communication. Within clubs and environments with large staff teams it was noted that "*it becomes difficult to ensure consistency of message and clarity of needs for each individual*" and "*very rarely do strength and conditioning professionals and coaches talk about periodization and training loads*".

Practitioners also identified the parent as a further practical challenge associated with the communication amongst stakeholders. Some practitioners identified that "*parents and administrators don't think this is important, all they want is immediate success*" and that parent "*education*" is important. Lastly, practitioners identified that motivation is a factor with "*the reality of motivating youth to participate in such activities is difficult*" and that "*an alarming percentage of youth have little to no connection with physical activity outside of school*".

## Discussion

To our knowledge, this study is the first to evaluate the knowledge, adherence, practices, and challenges of practitioners (from multiple roles, sports, and contexts) responsible for long-

term athletic development. Using a mixed methods approach, our novel findings demonstrated that practitioners generally; 1) recognise their responsibility for delivering long-term athletic development, 2) have a familiarity with existing developmental models, 3) have high adherence towards delivering long-term athletic development programmes underpinned by a focus upon health and wellbeing, and 4) identify a range of goals and realities of practices associated with long-term athletic development (i.e., it's for life, it's individually centered, and that a varied programme is key that is informed by monitoring). However, findings also suggested that practitioner's perceived definitions of athleticism and long-term athletic development are inconsistent, with psychological components (i.e., confidence and competence) often overlooked. Additionally, adherence to the NSCA pillars surrounding non-linear aspects of growth and maturation, monitoring and assessment, and the systematic progression and individualization of training programmes are limited. Furthermore, practitioners identified a range of challenges to the application of their long-term athletic development practices that suggest assistance is needed from a policy, practitioner, and participant level to produce positive outcomes associated with health, fitness, physical activity, and sporting performance for all youth.

## Knowledge and adherence

The main aims of long-term athletic development (i.e., influencing health and fitness, physical performance, confidence and competence in a range of movements, and reducing the relative risk of injury [1,7,8]) were corroborated by practitioners as the primary elements under their responsibility. It is encouraging that multiple practitioners (e.g., sport coaches, PE teachers) acknowledged their responsibility of delivering across these elements and appear to adhere to the NSCA Pillar 5 (i.e., health and wellbeing) within their programmes. However, whilst over 90% of practitioners identified their responsibility towards long-term athletic development components, a small proportion of practitioners suggested they were not accountable towards these aims. Whilst the contextual environments will differ between practitioners (e.g., a strength and conditioning coach working in a school *versus* a sport coach as part of a multidisciplinary support team within an academy programme), effective implementation of long-term athletic development for all youth necessitates *all stakeholders* recognise their responsibility to the cause. To achieve a unified approach, a clear and shared understanding of *athleticism* and *long-term athletic development* are required. To this point, practitioners' definitions of athleticism were mixed, and often unidimensional in nature predominantly focussed upon physical domains. For example, only 2.7% of practitioners were able to identify all four components (i.e., movement, physical, psychological, multiple environments) of athleticism [7], with just 2.0% including a psychological element (i.e., confidence and competence) suggesting a lack of appreciation for the multifactorial nature of the concept of athleticism and much attention on the physical qualities. Such findings are noteworthy, especially given the importance of developing confidence and perceived competence for the development of physical literacy (e.g., [33–36]) and other health-related behaviours [37], alongside the importance of psycho-social development for long-term sports participation and performance (e.g., [38,39]). Practitioners defined long-term athletic development as an extended process aimed at improving youth through appropriate planning and delivery of training, delivered within an appropriate environment resulting in multiple long-term outcomes. However, definitions were highly variable amongst practitioners with the content analysis demonstrating between 50–67% of responses supporting each content analysis theme. Furthermore, questions were raised as per a focus on the '*athlete*' over the '*individual*' suggesting some practitioners only focus upon athlete development rather than principles of long-term athletic development that apply to all youth as per international consensus [6] and position [7] statements. These findings suggest further work is

required to develop and promote a common consensus on long-term athletic development terminology amongst practitioners responsible for its successful implementation. Important considerations of the dynamic, complex and non-linear development of youth is required within practitioner education [2].

Practitioners reported greater familiarity with the LTAD and YPD models compared to the DMSP and NSCA Ten Pillars, with sport leaders, academics and strength and conditioning coaches having greater familiarity than other roles (e.g., PE teachers). This suggests that knowledge of the most recent information (e.g., NSCA Ten Pillars; [7]) may be limited compared to traditional models (e.g., LTAD model; [3]), or that practitioners may prefer the simplicity offered in the older models rather than the more recent iterations, which may also explain the contradictory definitions provided. This may occur if experienced practitioners (as per the current sample) rely on older models based on previous education and experiences. Furthermore, some practitioner groups not as well versed in athletic development or strength and conditioning (e.g., sport coaches, PE teachers) may require further education associated with these models to help enhance their awareness and understanding. Therefore, the dissemination of up-to-date and evidence-based models is required across all domains to ensure practitioners work towards a shared philosophy and practice of long-term athletic development.

Whilst practitioner's knowledge of long-term athletic development could be considered as contradictory, when practitioners were asked to rate their adherence to the NSCA Ten Pillars, scores were high. For example, adherence ratings across all pillars and practitioner roles had a median score of 4 or 5 (except for Pillar 9 for sport leaders and others, which may be due to their roles requiring limited contact time with participants). This suggests practitioners identified delivering on many of the recommended long-term athletic development practices suggested by Lloyd and colleagues [7]. However, considering increased obesity prevalence [21] and declining levels of motor skill competence [20], physical fitness [18] and physical activity [19] globally within youth populations suggests this is a key focus for explanation. For instance, the current participants may be delivering high quality practices (i.e., as reflected in their current practices reported in this study) but are not representative of the large scale and global population of practitioners responsible for long-term athletic development across all ages and stages. Alternately, practitioners may perceive their practices are better than they actually are, which may be related to social desirability bias, whereby individuals tend to present themselves in a favourable manner [40]. However, without further qualitative and observational research, adherence to the long-term athletic development principles is difficult to establish [41]. Therefore, whilst practitioners may rate their current practices quite highly, further research is required to understand if/how these practices align to long-term athletic development principles.

Adherence was generally high across all pillars, with Pillar 5 being the highest rated pillar. The focus upon this pillar is encouraging given the importance of promoting health and well-being and is supported by practitioners' descriptions of their current practice. However, like the previous conclusion regarding long-term athletic development practices, further investigation is required to explore how well practice reflects this principle. The lowest scoring pillars were Pillar 1 (highly individualized growth and development), 8 (monitoring and assessment) and 9 (systematically progress and individualize training programmes), which is noteworthy considering the dynamic and non-linear development of youth. These are the NSCA Pillars most associated with scientific principles of paediatric exercise and training science and thus may present the greatest challenge to implementation. Furthermore, research aligned to these Pillars is well established including growth and maturity (e.g., [42–44]), training load (e.g., [45–47]), fitness assessment (e.g., [48–50]) and appropriate implementation strategies (e.g., biobanding, [51]; strength & conditioning, [13]; movement competence, [52]). This suggests

that despite the quantity and robustness of existing research, further work is required for the appropriate translation, dissemination and education across multiple practitioners to promote research-informed practice within long-term athletic development pathways [53,54] and to support adherence, accuracy and consistency of planning, profiling and programme development for all youth.

## Practitioner perspectives: Practices and challenges

Practitioners described their current long-term athletic development practices and the challenges associated with them, which were thematically coded into two higher order themes of 1) goals; and 2) realities of delivering long-term athletic development. The first main theme within the goals was "it's for life", whereby practitioners identified their practices working towards a lifelong process with multiple outcomes (i.e., health, fitness, physical activity, sports performance) that were consistent with their adherence to Pillar 5. As such, it was evident practitioners felt responsible for the long-term development of individuals, to the extent that it was stated that it was their "*duty*". Closely related was the second theme, the individual centred journey, which aligned with philosophies and recommendations of the individual (participant or athlete) being central to all coaching practices [1,55]. To achieve an individual centred journey, three subthemes emerged: 1) fun and enjoyment, 2) developmental, and 3) within a safe and caring environment. These subthemes support research suggesting youth sports participation and training needs to be enjoyable [56,57], encourage competence and improvement in holistic development aspects (i.e., physical, technical, tactical, psycho-social) [58,59], and be delivered by those with genuine care for their participants [55,60,61].

The realities of delivering long-term athletic development centred upon 1) a varied programme, 2) monitoring development and 3) the practical challenges. A varied programme was associated with Pillars 2, 3, 4, 6 and 7 of the NSCA Ten Pillars [7] alongside other developmental models (e.g., DMSP, [4]). More specifically, practitioners reported their practices aligned with using multi-sport and -activity, movement development, and physical development alongside individualised practices and education as recently recommended [6–9]. To achieve this, multiple environments (e.g., PE session, after school clubs, through play, and sport specific sessions) were required with some sports clubs and schools having strength and conditioning provision [62] whereby others delivered through warm ups (e.g., RAMP warm up; [63,64]). Such practices demonstrate the need to have a curriculum underpinned by athleticism, offering multiple and varied opportunities for fun and engaging sessions within and outside the curriculum [1,8,14]. Furthermore, an education programme for enhancing knowledge within youth populations about the importance of fitness, health and other factors such as sleep, recovery and nutrition is required, and where possible these educational outcomes would be seamlessly integrated within the practical delivery [65].

Although the data suggested the Pillars associated with assessment and monitoring had lower adherence than others, monitoring was still identified as a theme, with practitioners emphasising its importance for growth and maturation [42,44], movement [66,67], fitness [68,69], and wellbeing and recovery [70,71]. Multiple methods of assessment were reported; however developing effective, valid and reliable methods across all domains of athletic and holistic development may be a challenge alongside the complications of collecting, analysing, interpreting, evaluating and presenting such information to inform decision making [72]. As such, practitioners questioned the use of traditional quantitative methods, instead suggesting the use of observational insights and the "coach's eye", a concept that is gaining traction in research settings [73,74]. However, recent research has also suggested coaches' judgements lack agreement with quantitative data [75], emphasising the need to confirm the validity of

this subjective information. This highlights the complexity of monitoring and assessment within youth populations, especially when constrained by facilities, equipment, and human resources. However, given the complexities of working with youth populations, it is imperative that practitioners apply available methods which are valid and reliable, to better understand the individual needs of each child. Utilising relationships with Universities, as seen in professional sport [76,77], is recommended as a positive way forward.

Whilst a range of positive practices were identified, multiple practical challenges towards delivering long-term athletic development was the third main theme, including eight lower order themes of governance and priorities, resources, education, early specialisation, high training volumes, staff communication, parents and youth motivation demonstrating the wide and complex challenges to overcome. Governance and priorities suggested challenges inconsistent with the goals of long-term athletic development, evident by the competitive nature of youth sports and competition between sports for participants and 'talent'. These challenges align with those identified in the early 2000s and a drive towards using sport for positive youth development (e.g., [78]). However, current findings suggest these problems may still apply, further emphasised by practitioners with a focus on winning and personal gain. As such, sporting organisations and their practitioners are encouraged to question the appropriateness of their pathways [79] and establish strategies to maximise positive youth development [80] for all associated long-term athletic development principles. These findings also align with the sub theme of education, where some practitioners failed to recognise their responsibility for long-term athletic development. This suggests more needs to be done to develop knowledge, philosophy and changing mindsets towards the purpose of youth sport alongside developing education provision to support the holistic development of youth practitioners.

Resources was a third practical challenge especially considering time availability with large groups of participants, which aligned to the lower adherence to the NSCA Pillar 9, related to individualisation of training. Whilst individualisation of training has been identified as important [6], considered alongside the non-linear, dynamic and complex development of youth [2,44], this is an understandable challenge when working with large groups. Instead of individualisation, it may be more appropriate to consider the concept of differentiation, as used within teaching (e.g., [81,82]), as a key principle to apply within long-term athletic development. Whilst this does not necessitate the development of session, weekly and annual plans for every individual the utilisation of varying practices to suit the needs of individuals through training prescription and session design should help overcome the resources challenge [83,84].

Whilst the findings demonstrated practitioners adhered to multi-sport and -activity practices, early sports specialisation was identified as a challenge, which is consistent with a plethora of research in the area (e.g., [2,8,85]). These challenges may be associated with multiple potential negative outcomes associated with early sports specialisation including increased risk of overuse injury [86], burnout [87], and blunting of motor skill development [88]. Whilst some sports and athletes may favour early specialisation, it is recommended that these individuals still participate in multi-sports and activities to develop a breadth of skills that may be needed at later timepoints with recent communication suggesting further evidence is required before we condemn it [89]. Aligned to early specialisation, a further challenge that was highlighted by responders was high training volumes. Paradoxically, while most youth fail to meet the recommended physical activity guidelines, some individuals, especially in youth sport, may undertake excessive workloads resulting in inadequate rest and recovery. Whilst this could be experienced by individuals who specialise in a single sport, other individuals may participate in multiple sports, within multiple environments delivered by multiple coaches resulting in what has been termed 'organised chaos' [45,90]. Arguably, this results from limited alignment and communication among sport stakeholders (including parents) in developing

appropriate programmes for the individual, with coach and programme-driven decisions usually the focus (as per coach priorities). This questions the individual centred journey identified as a key goal of long-term athletic development. Scantlebury and colleagues [46] have provided recommendations for managing such situations including enhanced communication, monitoring and collaboration; however, findings indicate an ongoing concern. Lastly, educating parents seems an important challenge to overcome [91,92] to provide appropriate programmes and support young people to be motivated to participate in long-term athletic development programmes.

## Strengths and limitations

To our knowledge, this study is the first to investigate the knowledge, adherence, practices, and challenges of practitioners' responsible for delivering long-term athletic development across youth populations. Using a mixed methods study design with a large sample of experienced practitioners across multiple roles who were responsible for long-term athletic development across multiple populations (i.e., sex, age, standard and sports). Such a strategy allowed a detailed, generalised and 'big picture' evaluation of the current landscape of optimising long-term athletic development. However, these strengths could also be acknowledged as limitations of the study, whereby a mixed method study utilising a wide and varied cohort of participants fails to acknowledge the intricacies of context within long-term athletic development practices within specific settings (e.g., secondary schools, football academies) across different nations and systems (unfortunately this information was not available). Furthermore, as practitioners were asked to self-rate their adherence (using a 5-point Likert scale) and describe their own practices, it is likely social desirability bias and score saturation may have occurred. Acknowledging these limitations, we believe this exploratory study provides a platform for future (and more context and role specific) work whilst demonstrating that long-term athletic development often occurs in multiple environments aligned to the philosophy of developing practices and recommendations for all youth. Furthermore, with the rapidly developing landscape of girl's sport and given the paucity of participants who worked specifically with females, a more targeted insight into LTAD practices and challenges for this demographic may support policy makers and practitioners to be dynamic in their design of LTAD in this developing sector of youth sport and physical activity.

## Conclusion

This study provided novel, mixed method, interesting and generalisable insights of the knowledge, adherence, practices and challenges of practitioners responsible for long-term athletic development of youth. In summary, practitioners recognised their responsibility for long-term athletic development outcomes (i.e., health and fitness, physical performance, confidence and competence to develop movement, and reduce injury risk), acknowledged high familiarity with existing developmental models, and reported high adherence towards delivering long-term athletic development programmes underpinned by a focus upon health and wellbeing. However, the definitions of athleticism and long-term athletic development were inconsistent and adherence towards some of the NSCA pillars (i.e., Pillars 1, 8 and 9) was limited. Furthermore, perceptions of long-term athletic development practices identified the goals and realities of delivering long-term athletic development should be a lifelong and individual centered journey delivered through a programme with high variety supported by relevant monitoring processes. However, multiple general challenges to delivering long-term athletic development were identified from the governance of youth sport, the resources available to deliver this, and the need for educating multiple stakeholders, which may be context and practitioner specific.

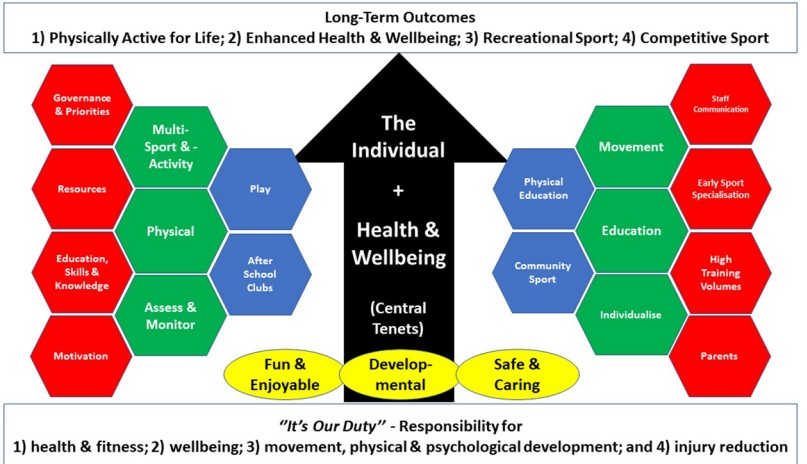

**Fig 1. Thematic map of the long-term athletic development practices and challenges.** Legend: White box = short- and long-term responsibilities, Black = it's for life with a focus on the individual and health and wellbeing, yellow = how this is achieved, Blue = where this is achieved; Green = the programme; Red = the practical challenge.

Whilst this is the first attempt to understand practitioners' knowledge, adherence, practices and challenges, a range of general recommendations can be provided for researchers and youth sport practitioners, particularly sports leaders. First, there is a need to enhance the fundamental knowledge of long-term athletic development for those working with youth through appropriate translation, dissemination and education of current knowledge and best practices. Second, a future research strategy for long-term athletic development should be orientated towards short- and long-term health and wellbeing given the reported importance of these areas identified in the study alongside the investigation of current practices to explore whether adherence to the principles of long-term athletic development are appropriate. Third, there is a need to accept and embrace the complexities and challenges of delivering long-term athletic development within multiple contexts and ascertain how current youth sport models alone may not be suitable to all populations. Lastly, to overcome these challenges multiple organisations and practitioners need to work collectively to offer the multiple potential benefits of long-term developmental outcomes focussed upon health and wellbeing that aim to deliver equal opportunities for all youth and work towards a fitter, healthier and more physically active population. Adopting the principles summarised in Fig 1 (i.e., a thematic map of the current findings) would provide clarity on the goals and realities of optimising long-term athletic development.

## Acknowledgments

This research was supported by the United Kingdom Strength and Conditioning Association and we would like to thank them for their support on participant recruitment. We would also like to thank and acknowledge all the practitioners who took their time to offer their valuable insights by completing the survey.

## Author Contributions

**Conceptualization:** Kevin Till, Rhodri S. Lloyd, Graham Williams, Joey C. Eisenmann.

**Data curation:** Kevin Till, Sam McCormack, Joey C. Eisenmann.

**Formal analysis:** Kevin Till, Sam McCormack.

**Methodology:** Kevin Till, Rhodri S. Lloyd, Graham Williams, Joey C. Eisenmann.

**Writing – original draft:** Kevin Till, Rhodri S. Lloyd, Joseph Baker, Joey C. Eisenmann.

**Writing – review & editing:** Kevin Till, Rhodri S. Lloyd, Sam McCormack, Graham Williams, Joseph Baker, Joey C. Eisenmann.

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
