## [Decision Letter · Decision Letter 0]

9 Nov 2021

PONE-D-21-12094Optimising long-term athletic development: An investigation of practitioners’ knowledge, adherence, practices and challengesPLOS ONE

Dear Dr. Till,

Thank you for submitting your manuscript to PLOS ONE. After careful consideration, we feel that it has merit but does not fully meet PLOS ONE’s publication criteria as it currently stands. Therefore, we invite you to submit a revised version of the manuscript that addresses the points raised during the review process. Both reviewers appreciated your manuscript however they noted some minor remarks Please submit your revised manuscript by Dec 24 2021 11:59PM. If you will need more time than this to complete your revisions, please reply to this message or contact the journal office at plosone@plos.org. Please include the following items when submitting your revised manuscript:A rebuttal letter that responds to each point raised by the academic editor and reviewer(s). You should upload this letter as a separate file labeled 'Response to Reviewers'.A marked-up copy of your manuscript that highlights changes made to the original version. You should upload this as a separate file labeled 'Revised Manuscript with Track Changes'.An unmarked version of your revised paper without tracked changes. You should upload this as a separate file labeled 'Manuscript'.If applicable, we recommend that you deposit your laboratory protocols in protocols.io to enhance the reproducibility of your results. Protocols.io assigns your protocol its own identifier (DOI) so that it can be cited independently in the future. For instructions see: https://journals.plos.org/plosone/s/submission-guidelines#loc-laboratory-protocols. Additionally, PLOS ONE offers an option for publishing peer-reviewed Lab Protocol articles, which describe protocols hosted on protocols.io. Read more information on sharing protocols at https://plos.org/protocols?utm_medium=editorial-email&utm_source=authorletters&utm_campaign=protocols.

We look forward to receiving your revised manuscript.

Kind regards,

Maria Francesca Piacentini

Academic Editor

PLOS ONE

Journal Requirements:

3. Your abstract cannot contain citations. Please only include citations in the body text of the manuscript, and ensure that they remain in ascending numerical order on first mention.

Reviewers' comments:

Reviewer's Responses to Questions

**Comments to the Author**

1. Is the manuscript technically sound, and do the data support the conclusions?

Reviewer #1: Yes

Reviewer #2: Yes

2. Has the statistical analysis been performed appropriately and rigorously? 

Reviewer #1: Yes

Reviewer #2: Yes

3. Have the authors made all data underlying the findings in their manuscript fully available?

Reviewer #1: No

Reviewer #2: Yes

4. Is the manuscript presented in an intelligible fashion and written in standard English?

Reviewer #1: Yes

Reviewer #2: Yes

5. Review Comments to the Author

Reviewer #1: The study delivered a mixed-method questionnaire to more than 200 practitioners to evaluate the knowledge, adherence, practices, and challenges of practitioners responsible for delivering long-term athletic development. The manuscript is well written and the methods are efficiently reported. The results are reported with many details; this makes the manuscript quite long. However, the conclusions and the summarizing figure makes the core finding clearly identifiable. The results are interpreted correctly and discussed in light of the updated literature. For these reasons, I want to congratulate the Authors on this study.

I only have one major comment (easily amendable): the national distribution of the practitioners should be included as the nationality may affect their sport-specific education. The results should be discussed accordingly.

Few minor comments:

• Line 128 – while it is clear to me, it may be useful to explicit what “A-Levels / BTEC” means for stranger readers.

• Was the questionnaire delivered only in English? Please include this information.

Reviewer #2: Interesting study that explores the ''knowledge, adherence, practices, and challenges of practitioners

responsible for delivering long-term athletic development.'' This study may be the only that went that far in documenting adherence and practices. This was realized with an appropriate methodological approach (i.e. mixed). Well written.

Page 6: The sample is heterogeneous. It is surprising that they are all participants considered as ''practitionners responsible for delivering long-term athletic development''. Like parents?

The description of the sample should be upgraded. In the discussion, a contextualization of the sample should be included to better explain what are the implicaions in terms of generalization of findings/conclusions/recommendations.

In the different sports represented you mention hockey, is it field or ice?

Page 10 Table 2: Definition 3 and 4 seem redundant

I am a little bit surprised that social desirability was not mentionned in the ''Strengths and Limitations'' section. When professionals are asked about their respect of best practices, they tend to self-score higher. Self-reporting of practice,

may be subject to social desirability bias. Nobody is against virtue. However, I don't believe that it may affect comparison of adherence between pillars. It may explain however, the saturation of the scores (table 5). Also to what extent do these strongly adherent practitionners really apply what they preach in their practice? With some of them you were able to obtain concrete examples of action (e.g. p21) but with most only general statements were declared.

Page 15: Also, why is there a gap between sport leaders and other ''practitionners'' ? Isn't it worrisome that sport leaders don't understand that progression and individualisation are essential for long-term development.

On page 32 it is written that : ''This study is the first to investigate the knowledge, adherence, practices, and challenges of

practitioners’ responsible for delivering long-term athletic development.'' It is not true. Indeed, I am surprised that the research on the Canadian version of LTAD seems to have been evacuated. The following references have information that are relevant for the present study and investigated ''the knowledge, adherence, practices, and challenges of PTAD delivery:

-Beaudoin, C., Callary, B., & Trudeau, F. (2015). Coaches’ adoption and implementation of Sport Canada’s long-term athlete development model. SAGE Open, 5(3), 2158244015595269.

-Banack, H. R., Bloom, G. A., & Falcão, W. R. (2012). Promoting long term athlete development in cross country skiing through competency-based coach education: A qualitative study. International Journal of Sports Science & Coaching, 7(2), 301-316.

-Black, D.E., et Nicholas L. Holt, N.L. (2009). Athlete development in ski racing: perceptions of coaches and parents. International Journal of Sports Science & Coaching, 4(2), 245-260.

-Chevrier J., Roy M., Turcotte S., Culver D.M., Cybulski S. (2016). Skills trained by coaches of Canadian male volleyball teams: A comparison with long-term athlete development guidelines. International Journal of Sports Science & Coaching, 11(3), 410-421.

-Frankish, M. T., Beaudoin, C., & Callary, B. (2012). Cross-Country ski coaches and the LTAD model: Exploring attributes of adoption. Revue phénEPS/PHEnex Journal, 4(2). https://ojs.acadiau.ca/index.php/phenex/article/view/1458

-Jurbala, P., & Stevens, J. (2020). A whole new ballgame: an analysis of the context and adoption of long-term athlete development in community sport. Managing Sport and Leisure, 1-17.

-Millar, P., Clutterbuck, R., & Doherty, A. (2020). Understanding the adoption of long-term athlete development in one community sport club. Managing Sport and Leisure, 25(4), 259-274.

6. PLOS authors have the option to publish the peer review history of their article (what does this mean?). If published, this will include your full peer review and any attached files.

Reviewer #1: No

Reviewer #2: No

---

## [Author Response · Author response to Decision Letter 0]

18 Nov 2021

Please note all line numbers refer to the tracked change document

Reviewer #1: 

The study delivered a mixed-method questionnaire to more than 200 practitioners to evaluate the knowledge, adherence, practices, and challenges of practitioners responsible for delivering long-term athletic development. The manuscript is well written and the methods are efficiently reported. The results are reported with many details; this makes the manuscript quite long. However, the conclusions and the summarizing figure makes the core finding clearly identifiable. The results are interpreted correctly and discussed in light of the updated literature. For these reasons, I want to congratulate the Authors on this study.

Thank you very much for your positive comments. 

I only have one major comment (easily amendable): the national distribution of the practitioners should be included as the nationality may affect their sport-specific education. The results should be discussed accordingly.

This is a really interesting point. Although this would seem easily amendable, unfortunately, we could not provide this level of information within the paper. While our survey provided latitude and longitude destinations for participants, the survey (using Qualtrics) only collected this information for 130 of our 236 participants (I don’t know why this was the case). Therefore, I don’t think it is appropriate to report the nationality based on the limited information available. I have included this as a limitation on lines 758-759

Few minor comments:

• Line 128 – while it is clear to me, it may be useful to explicit what “A-Levels / BTEC” means for stranger readers.

I have added the meaning of A-Levels and BTECs on lie 131-132

• Was the questionnaire delivered only in English? Please include this information.

In English only has been added to line 149 

Reviewer #2: 

Interesting study that explores the ''knowledge, adherence, practices, and challenges of practitioners responsible for delivering long-term athletic development.'' This study may be the only that went that far in documenting adherence and practices. This was realized with an appropriate methodological approach (i.e. mixed). Well written.

Thank you for your positive comments and suggestions below. We have addressed each comment below. 

Page 6: The sample is heterogeneous. It is surprising that they are all participants considered as ''practitionners responsible for delivering long-term athletic development''. Like parents?

We agree the sample is heterogenous, which provides a strength and weakness of the study (see study strengths and limitations). We have removed parents from line 126 as we agree they were not a practitioner. All data has remained within the study as although two participants identified their primary role as a parent they also had a secondary ‘practitioner’ role. 

The description of the sample should be upgraded. In the discussion, a contextualization of the sample should be included to better explain what are the implications in terms of generalization of findings/conclusions/recommendations.

We have presented the data that we collected and are unsure how to upgrade the description of the sample further. However, we acknowledge that we could present the interactions between the sample but every participant had a different range of qualifications, experiences and worked in multiple roles and contexts making the presentation of such information difficult and likely confusing. However, we have updated the discussion, strengths and limitations and conclusion sections to re-emphasise the generalisation of the findings in relation to the broad sample used. 

In the different sports represented you mention hockey, is it field or ice?

It was actually both field and ice hockey. This has been added on line 135. 

Page 10 Table 2: Definition 3 and 4 seem redundant

Thank you for this comment. While we acknowledge your points, the definitions are used to show examples of practitioners’ definitions of athleticism. Definition 3 and 4 are similar but definition 4 states in a range of physical activities aligned to the multiple environments part of the definition (see table below). As such we would respectfully prefer to keep the definitions within the table. 

3 (33) 15.1% ‘’one’s ability to perform a broad range of movements, with confidence and competence, to develop a range of physical qualities including strength, speed, power, agility and endurance’’

4 (3) 2.7% ‘’the ability to participate confidently and competently in a range of physical activities and in doing so demonstrate flow, strength, speed and coordinated range of motion’’

I am a little bit surprised that social desirability was not mentionned in the ''Strengths and Limitations'' section. When professionals are asked about their respect of best practices, they tend to self-score higher. Self-reporting of practice, may be subject to social desirability bias. Nobody is against virtue. However, I don't believe that it may affect comparison of adherence between pillars. It may explain however, the saturation of the scores (table 5). Also to what extent do these strongly adherent practitionners really apply what they preach in their practice? With some of them you were able to obtain concrete examples of action (e.g. p21) but with most only general statements were declared.

This is a really interesting comment. We have now referred to social desirability bias on lines 626-628. We have also acknowledged social desirability and score saturation as a study limitation (Lines 759-761). 

In relation to your point re practising what they preach, we refer to this within lines 627-631 to provide examples of where further research is required to understand this. 

Page 15: Also, why is there a gap between sport leaders and other ''practitionners'' ? Isn't it worrisome that sport leaders don't understand that progression and individualisation are essential for long-term development.

While these differences were not significant we did highlight that sport leaders and others only scored a 3 for Pillar 9 (progress and individualise training) on Line 617. This question was in relation to their practices and we have added an explanation on line 617-618 why this may be the case. 

On page 32 it is written that : ''This study is the first to investigate the knowledge, adherence, practices, and challenges of practitioners’ responsible for delivering long-term athletic development.'' It is not true. Indeed, I am surprised that the research on the Canadian version of LTAD seems to have been evacuated. The following references have information that are relevant for the present study and investigated ''the knowledge, adherence, practices, and challenges of PTAD delivery:

Thank you for this comment. There seems some difference here between the Canadian long-term athlete development model (as per the references below) and the focus on “long-term athletic development”, building on the NSCA position statement (Lloyd et al., 2016). We introduce these concepts in the introduction (paragraph 1) and the move away from a ‘talent only’ strategy as proposed by existing models. Therefore, while studies may have examined LTAD practices of coaches (as per your list below), we are not aware of any study that has explored the practices related to long-term athletic development according to the definition of Lloyd et al., (2016) “habitual development of athleticism over time to improve health and fitness, enhance physical performance, reduce the relative risk of injury, and develop the confidence and competence of all youth”.

As such we have tried to emphasise this point in the introduction and include some of the references you provided below within the introduction. Please see lines 83-84. 

-Beaudoin, C., Callary, B., & Trudeau, F. (2015). Coaches’ adoption and implementation of Sport Canada’s long-term athlete development model. SAGE Open, 5(3), 2158244015595269.

-Banack, H. R., Bloom, G. A., & Falcão, W. R. (2012). Promoting long term athlete development in cross country skiing through competency-based coach education: A qualitative study. International Journal of Sports Science & Coaching, 7(2), 301-316.

-Black, D.E., et Nicholas L. Holt, N.L. (2009). Athlete development in ski racing: perceptions of coaches and parents. International Journal of Sports Science & Coaching, 4(2), 245-260.

-Chevrier J., Roy M., Turcotte S., Culver D.M., Cybulski S. (2016). Skills trained by coaches of Canadian male volleyball teams: A comparison with long-term athlete development guidelines. International Journal of Sports Science & Coaching, 11(3), 410-421.

-Frankish, M. T., Beaudoin, C., & Callary, B. (2012). Cross-Country ski coaches and the LTAD model: Exploring attributes of adoption. Revue phénEPS/PHEnex Journal, 4(2). https://ojs.acadiau.ca/index.php/phenex/article/view/1458

-Jurbala, P., & Stevens, J. (2020). A whole new ballgame: an analysis of the context and adoption of long-term athlete development in community sport. Managing Sport and Leisure, 1-17.

-Millar, P., Clutterbuck, R., & Doherty, A. (2020). Understanding the adoption of long-term athlete development in one community sport club. Managing Sport and Leisure, 25(4), 259-274.

---

## [Decision Letter · Decision Letter 1]

11 Jan 2022

Optimising long-term athletic development: An investigation of practitioners’ knowledge, adherence, practices and challenges

PONE-D-21-12094R1

Dear Dr. Till,

We’re pleased to inform you that your manuscript has been judged scientifically suitable for publication and will be formally accepted for publication once it meets all outstanding technical requirements.

Kind regards,

Maria Francesca Piacentini

Academic Editor

PLOS ONE

Additional Editor Comments (optional):

Reviewers' comments:

Reviewer's Responses to Questions

**Comments to the Author**

1. If the authors have adequately addressed your comments raised in a previous round of review and you feel that this manuscript is now acceptable for publication, you may indicate that here to bypass the “Comments to the Author” section, enter your conflict of interest statement in the “Confidential to Editor” section, and submit your "Accept" recommendation.

Reviewer #1: All comments have been addressed

Reviewer #2: All comments have been addressed

2. Is the manuscript technically sound, and do the data support the conclusions?

Reviewer #1: Yes

Reviewer #2: Yes

3. Has the statistical analysis been performed appropriately and rigorously? 

Reviewer #1: Yes

Reviewer #2: Yes

4. Have the authors made all data underlying the findings in their manuscript fully available?

Reviewer #1: Yes

Reviewer #2: Yes

5. Is the manuscript presented in an intelligible fashion and written in standard English?

Reviewer #1: Yes

Reviewer #2: Yes

6. Review Comments to the Author

Reviewer #1: (No Response)

Reviewer #2: You addressed my concerns and questions to my satisfaction. It will be an interesting paper. The limits and strengths of manuscript are better detailed

7. PLOS authors have the option to publish the peer review history of their article (what does this mean?). If published, this will include your full peer review and any attached files.

Reviewer #1: No

Reviewer #2: No

---

## [Editor Report · Acceptance letter]

14 Jan 2022

PONE-D-21-12094R1 

Optimising long-term athletic development: An investigation of practitioners’ knowledge, adherence, practices and challenges 

Dear Dr. Till:

I'm pleased to inform you that your manuscript has been deemed suitable for publication in PLOS ONE. Congratulations! Your manuscript is now with our production department. 

Kind regards, 

on behalf of

Dr. Maria Francesca Piacentini 

Academic Editor

PLOS ONE